

# Multimodel assessments of human and climate impacts on mean annual streamflow in China

Xingcai Liu[1,2], Wenfeng Liu[2,3], Hong Yang[2,4], Qiuhong Tang[1,5], Martina Flörke[6], Yoshimitsu Masaki[7], Hannes Müller Schmied[8,9], Sebastian Ostberg[10], Yadu Pokhrel[11], Yusuke Satoh[12,13], Yoshihide Wada[12]

[1]Key Laboratory of Water Cycle and Related Land Surface Processes, Institute of Geographic Sciences and Natural Resources Research, Chinese Academy of Sciences, A11, Datun Road, Chaoyang District, Beijing, China
[2]Eawag, Swiss Federal Institute of Aquatic Science and Technology, Ueberlandstrasse 133, CH-8600 Duebendorf, Switzerland
[3]Laboratoire des Sciences du Climat et de l'Environnement, LSCE/IPSL, CEA-CNRS-UVSQ, Université Paris-Saclay, F-91191 Gif-sur-Yvette, France
[4]Department of Environmental Sciences, MGU, University of Basel, Petersplatz 1, CH-4003 Basel, Switzerland
[5]College of Resources and Environment, University of Chinese Academy of Sciences, Beijing 100049, China
[6]Center for Environmental Systems Research, University of Kassel, Kassel, Germany
[7]Graduate School of Science and Technology, Hirosaki University, Hirosaki, Japan
[8]Institute of Physical Geography, Goethe-University Frankfurt, Altenhöferallee 1, 60438 Frankfurt, Germany
[9]Senckenberg Biodiversity and Climate Research Centre (SBiK-F), Senckenberganlage 25, 60325 Frankfurt, Germany
[10]Earth System Analysis, Potsdam Institute for Climate Impact Research (PIK), Potsdam, Germany
[11]Department of Civil and Environmental Engineering, Michigan State University, East Lansing, MI 48824 United States of America
[12]International Institute for Applied Systems Analysis, Laxenburg, Austria
[13]National Institute for Environmental Study, Tsukuba, Japan

*Correspondence to*: Qiuhong Tang (tangqh@igsnrr.ac.cn)

**Abstract.** Human activities, as well as climate change, have had increasing impacts on natural hydrological systems, particularly streamflow. However, quantitative assessments of these impacts are lacking on large scales. In this study, we use the simulations from six global hydrological models driven by three meteorological forcings to investigate direct human impact (DHI) and climate change impact on streamflow in China. Results show that, in the sub-periods of 1971-1990 and 1991-2010, one-fifth to one-third of mean annual streamflow (MAF) reduced due to DHI in northern basins and much smaller (< 4%) MAF reduced in southern basins. From 1971-1990 to 1991-2010, total MAF changes range from -13% to 10% across basins, wherein the relative contributions of DHI change and climate change show distinct spatial patterns. DHI change caused decreases in MAF in 70% of river segments, but climate change dominated the total MAF changes in 88% of river segments of China. In most northern basins, climate change results in changes of -9% to 18% of MAF, while DHI change results in decreases of 2% to 8% in MAF. In contrast with the impacts of climate change that may increase or decrease streamflow, DHI change almost always contributes to decreases in MAF over time, wherein water withdrawals are supposed to be the major impact on streamflow. This quantitative assessment can be a reference for



attribution of streamflow changes at large scales despite uncertainty remains. We highlight the significant DHI in northern basins and the necessity to modulate DHI through improved water management towards a better adaptation to future climate change.

Keywords: streamflow; human impact; multimodel simulation; ISIMIP2a; China

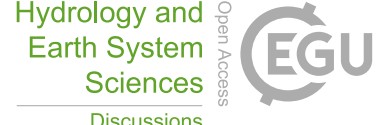



# 1 Introduction

Human activities have remarkably intensified and significantly altered hydrological regimes and water resources worldwide (Oki and Kanae, 2006; Döll et al. 2009; Tang and Oki, 2016). They have been reported to have aggravated hydrological drought and impaired hydrological resilience in many regions (Wada et al., 2013; Wada and Heinrich, 2013; Veldkamp et al., 2017). Human impact (here we only consider the direct human impact (DHI), e.g., that caused by the construction and management of dams and reservoirs, water withdrawal from surface water, and groundwater pumping, etc.) on streamflow has been on the rise across the world (Jaramillo and Destouni, 2015), causing the same order of magnitude of hydrologic alterations as by climate change in some regions (Ian and Reed, 2012; Haddeland et al., 2014; Zhou et al., 2015). As such, there has been increased attention in attributing hydrological impacts from various drivers (Patterson et al., 2013; Tan and Gan, 2015; Bosmans et al., 2017). Understanding the relative contributions of DHI to streamflow changes is of great importance for climate change adaptation and sustainable development (Yin et al., 2017).

In China, the hydrological system is experiencing significant changes induced by both climate change and human activities (Piao et al., 2010; Tang et al., 2013; Liu et al., 2014; Wada et al., 2017). Great efforts have been made to quantify the relative contributions of DHI in China (Liu and Du, 2017). Some studies have shown that DHI outweighed climatic impact on streamflow / runoff in several small catchments in the Hai River (Wang et al., 2009; Wang et al., 2013b) and the Yellow River (Li et al., 2007; Tang et al., 2008; Zhan et al., 2014; Chang et al., 2016). Other studies have reported that the construction and operation of the Three Gorges Reservoir resulted in considerable changes in streamflow (Wang et al., 2013a) but DHI contributed to small changes in streamflow in some catchments (Liu et al., 2012; Ye et al., 2013) and slight changes in lake areas in the Yangtze River basin (Wang et al., 2017). Most of these studies attributed human impact by comparing observed streamflow to simulations which were estimated with a climate elasticity approach based on the Budyko framework (Zhang et al., 2001) or with hydrological models (Wang et al., 2009; Wang et al., 2010; Yuan et al., 2018). These assessments largely relied on hydroclimatic observations and were performed on relatively small catchment scales to obtain quantitatively distinguishable attributions. The previous studies assessed DHI on streamflow changes at the outlets of catchments, but the spatial extents of the impacts have not been adequately examined. As mentioned above, many previous studies reported large DHI on streamflow; however, a recent large-scale assessment over the United States and Canada showed that human activities such as water management did not substantially alter the hydrological effects of climate change (Ficklin et al., 2018). In addition, the potential uncertainty associated with DHI and streamflow simulations can hardly be estimated from a single model assessment as done in previous studies. Therefore, an improved assessment with larger spatial coverage and by employing a multimodel comparison approach is essential to understand regional difference and associated uncertainty of the impacts.

The recent development of human impact parameterizations in hydrological models has facilitated the assessment of the DHI on streamflow (Pokhrel et al., 2016; Liu et al., 2017b; Veldkamp et al, 2018). Consequently, several global hydrological



modelling initiatives considering human impact have been undertaken, e.g., by the Inter-Sectoral Impact Model Intercomparison Project phase 2a (ISIMIP2a, Gosling et al., 2017). Under the ISIMIP2a framework, retrospective simulations of hydrological changes were performed for both natural conditions and those with human activities by six global hydrological models (GHMs). The simulations provide a basis for quantifying the streamflow changes caused by various drivers in a consistent manner on large scales. Meanwhile, the grid-based simulations allow an attribution at different geographic levels and, therefore, provide more detail information about regional streamflow changes. The ISIMIP2a simulations have included the most important DHI over large scales including the operation of reservoirs/dams on rivers as well as sectoral water withdrawals for irrigation, industry, domestic, and livestock. In this study, using the ISIMIP2a multimodel simulations, we quantify the relative contribution of DHI and climate change on streamflow changes in the major river basins in China at decadal timescale during the 1971-2010 period. This is the first study to perform such a quantitative assessment for all rivers of China with comparable modeling experiments. This study can serve as a reference for attribution of streamflow changes at large scales that can facilitate regional water resources management under climate change and growing human impact on freshwater system.

## 2 Method and data

### 2.1 Simulation data

In this study, we use the simulations of monthly streamflow of China produced by six GHMs, namely, DBH (Tang et al., 2007, 2008; Liu et al., 2016), H08 (Hanasaki et al., 2008a, 2008b), LPJmL (Bondeau et al. 2007; Rost et al. 2008; Biemans et al. 2011; Schaphoff et al. 2013), MATSIRO (Takata et al., 2003; Pokhrel et al., 2015), PCR-GLOBWB (Wada et al., 2014), WaterGAP2 (Flörke et al., 2013; Müller Schmied et al., 2014, 2016). Two experiments, i.e., simulations with (VARSOC) and without (NOSOC) human impact, were performed at a half-degree spatial resolution for the 1971-2010 period by using the six GHMs following the ISIMIP2a simulation protocol (https://www.isimip.org/protocol/#isimip2a). All the model runs used the same river routing map (DDM30, Döll and Lehner, 2002). Human impact considered in the VARSOC experiment (see the maps in Figure S1 and Table S1 for more details) include the time-varying areas for both irrigated and rainfed cropland (Fader et al., 2010; Portmann et al., 2010) and reservoirs (dams) from the Global Reservoir and Dam (GRanD) Database (Lehner et al., 2011) including their commissioning year (see Figure S1 and Table S1 for more detail). Reservoir regulation was considered in the VARSOC experiment, which often reduces high streamflow in high-flow seasons and increases streamflow in dry seasons (Masaki, et al., 2017). Inter-basin water transfer was not considered in any of the model runs. For both experiments, the GHMs were forced by three global meteorological forcing products (GMFs), i.e., the PGMFD v.2 (Princeton) (Sheffield et al., 2006), GSWP3 (http://hydro.iis.u-tokyo.ac.jp/GSWP3/), and a combination of WFD (until 1978, Weedon et al., 2011) and WFDEI (from 1979 onwards, Weedon et al., 2014) datasets. Ensembles of annual streamflow are derived from the simulations of NOSOC (referred to as $Q_n$) and VARSOC (referred to as $Q_v$) experiments, respectively, for river segments (here a grid cell is treated as a river segment regardless of the cases that a grid



cell contains several small river segments) which are then spatially averaged for individual basins. Long-term mean annual streamflow (MAF) in each river segment is calculated for both NOSOC and VARSOC simulations over a specific period (see section 2.3) and then is spatially-averaged over individual basins for each ensemble member. In addition to streamflow, total runoff from NOSOC and VARSOC simulations and water withdrawals from VARSOC simulations are also derived at

5 grid cells and individual basins for associated analyses. The simulations may have large uncertainties over the Tibetan Plateau because long-term meteorological and streamflow observations are sparse in this region (Zhang et al., 2017). Therefore, the simulation data in the Tibetan Plateau region are removed and are not included in spatial averages by masking the grid cells with altitudes higher than 4000 meters in all analyses.

## 2.2 Observed monthly streamflow and reported water withdrawals

The ISIMIP2a streamflow simulations have been extensively validated with observations over the world in several studies (Liu et al, 2017b; Veldkamp et al., 2018; Zaherpour et al., 2018), but were not fully evaluated in China due to limited observations, particularly for the water withdrawals. Therefore, before the quantitative attribution, a preliminary evaluation of the multimodel simulations is performed, which may add confidence regarding the GHMs' performance over China. Observations of monthly streamflow from 44 hydrological stations in China (Figure 1) during 1971-2000 are used for model

validation. The observations since 2001 are not available in this study. Some stations are relocated on the map to reconcile the catchment areas of the stations and the accumulative flow areas of corresponding gird cells from the DDM30 river network. After relocation, the differences are mostly less than 10% (about 50% at 5 stations) between the reported catchment areas of stations and the accumulative flow areas from the DDM30 river network. Annual water withdrawals in individual basins for the years of 1980, 1985, 1990, 1995, and 1997-2010 were collected from China Water Resources Bulletin from

the Ministry of Water Resources (MWR) of China (http://www.mwr.gov.cn/sj/tjgb/szygb/).

## 2.3 Streamflow changes and attribution

The study period is evenly split into two sub-periods (P1 for 1971-1990 and P2 for 1991-2010). The DHI-induced MAF changes over time is calculated as:

$$\begin{cases} Q_h^{P1} = 100 \times \frac{Q_v^{P1} - Q_n^{P1}}{Q_n^{P1}} \\ Q_h^{P2} = 100 \times \frac{Q_v^{P2} - Q_n^{P2}}{Q_n^{P2}} \end{cases} \tag{1}$$

where $Q_h^{P1}$ and $Q_h^{P2}$ denote MAF changes (%) induced by DHI during the sub-periods P1 and P2, respectively; $Q_v^{P1}$ and $Q_v^{P2}$ denote MAF from the VARSOC experiment for the two sub-periods, respectively; $Q_n^{P1}$ and $Q_n^{P2}$ denote MAF from the NOSOC experiment for the two sub-periods, respectively.





The contribution of *DHI change* (corresponding to *climate change*) on streamflow changes between the two sub-periods is also examined. MAF difference between the two periods in the VARSOC experiment is defined as the total MAF changes ($\Delta Q_a$) caused by both climate change and DHI change from P1 to P2, which is expressed as a percentage of the MAF of the first sub-period P1:

$$\Delta Q_a = 100 \times \frac{Q_v^{P2} - Q_v^{P1}}{Q_v^{p1}}.$$ 

(2)

The difference between the two periods in the NOSOC experiment is defined as streamflow changes induced by only climate change ($\Delta Q_c$) and expressed as a percentage of $Q_v^{P1}$ in order to be comparable with $\Delta Q_a$:

$$\Delta Q_c = 100 \times \frac{Q_n^{P2} - Q_n^{P1}}{Q_v^{p1}}.$$ 

(3)

The difference between $\Delta Q_a$ and $\Delta Q_c$ then counts as MAF changes induced by DHI change ($\Delta Q_h$) between the two sub-periods:

$$\Delta Q_h = \Delta Q_a - \Delta Q_c = 100 \times \frac{(Q_v^{P2} - Q_v^{P1}) - (Q_n^{P2} - Q_n^{P1})}{Q_v^{p1}}.$$ 

(4)

Unless otherwise stated, $\Delta Q_a$, $\Delta Q_c$, $\Delta Q_h$ are relative changes (%) with respect to $Q_v^{P1}$ in this paper.

To address the potential uncertainty resulting from the use of sub-periods, similar analyses are performed for three different sub-periods, namely, 1981-1990, 1991-2000, and 2001-2010, with comparison to the sub-period 1971-1980. For these analyses, MAF is calculated over each decade.

In addition to streamflow, changes in water withdrawals and total runoff between the two sup-periods are also analyzed to explore their links with MAF changes.

**2.4 Multimodel ensemble**

Ensemble medians across the 18 GHM-GMF combinations (6 GHMs and 3 GMFs) are used for analyses of streamflow and runoff. But 12 ensemble members are used for water withdrawals because only 4 GHMs (H08, LPJmL, PCR-GLOBWB, and MATSIRO) provide related output for the ISIMIP2a simulations. The interquartile range (IQR), i.e., the range between 25th and 75th percentiles, is calculated to present the spread across multimodel ensembles. The ratio of IQR to the median is used to measure the uncertainty in multimodel simulations of streamflow, which is comparable across regions.



## 3 Results

### 3.1 Evaluation of multimodel simulations

In this study, the northern basins refer to Songhua River (SH), Liao River (LR), Northwest Rivers (NW), Hai River (HA), Yellow River (YR), Huai River (HU); and the southern basins refer to the Yangtze River (YZ), Southeast Rivers (SE), Southwest Rivers (SW), Pearl River (PR) (Figure 1). The ensemble medians of MAF at grid cells over the 1971-2000 period from the VARSOC experiment show distinct spatial pattern of high streamflow in southern basins and relatively low streamflow in northern basins (Figure 1). The multimodel simulations show larger spreads in the northern basins. The ratios of IQR/median are larger than 1 or 2 in the Northwest basin, the Yellow River basin and Liao River basin. Smaller spread (IQR/median less than 0.5) is found in the middle and lower reaches of the Yangtze River basin, the Pearl River basin, and the Southeast basin. The inner plot shows the comparison between observed seasonal streamflow averaged across all hydrological stations and the averaged simulations in all river segments identified by stations over the 1971-2000 period. The ensemble medians of seasonal cycle generally coincide with the observations. However, there are large variations across all model ensembles with some of them deviating from observations. It should be noted that the stations are located at different reaches of individual basins. Thus, the station-averaged seasonality is largely dominated by those with large streamflow (e.g., at the lower reaches). The model spreads in the ensembles of seasonal streamflow in the northern basins are usually larger than those in the southern basins (see Figure S2 for each basin), which underlines the necessity of using ensemble medians rather than individual models for the attribution of streamflow changes.

Compared to the reported data by the Ministry of Water Resources (MWR) of China, the ensemble medians from ISIMIP2a simulations underestimated water withdrawals in most northern basins except for the Yellow River (Figure 2). The simulations underestimate water withdrawals by more than 50% in the Northwest Rivers and the Hai River and by more than 30% in the Songhua River and Liao River. The simulated water withdrawals are 12% less than reported data in the Huai River. In the Yellow River and the Southeast Rivers, water withdrawals are overestimated by 20% or more. The overestimation of water withdrawals is the largest (80%) in the Southwest Rivers. Small differences between simulations and reported data are found in the Yangtze River (-1%) and the Pearl River (-6%).

### 3.2 Annual streamflow and DHI-induced streamflow change

Figure 3 shows the spatially averaged ensemble medians of $Q_n$ and $Q_v$ over China, the northern and southern basins. $Q_n$ and $Q_v$ show considerable annual variations and no statistically significant trends over the 1971-2010 period. The relative differences between $Q_n$ and $Q_v$ over China range from -8% to -4% and show a statistically significant downward trend over the study period (Figure 3a). The differences between $Q_n$ and $Q_v$ over the northern basins are larger than those for the southern basins. The absolute differences (not shown here) are -37 to -14 ($m^3/s$) for the northern basins and are -37 to -7 ($m^3/s$) for the southern basins. The relative differences for the northern basins (Figure 3b) are also larger than those for the



southern basins (Figure 3c). The former ranges from -30% to -10% while the latter ranges from -4% to -1%. Statistically significant downward trend is found in the relative differences for the northern basins, while non-significant downward trend is found for the southern basins. The downward trend in the differences indicates that annual streamflow has been increasingly affected by human impact.

**3.3 MAF altered by DHI in the two sub-periods**

Considerable decreases in long-term MAF are induced by DHI in the two sub-periods (Figure 4a and 4b for $Q_h^{P1}$ and $Q_h^{P2}$, respectively) in most northern basins. About 3% and 4% of total river segments in China show large negative values (i.e., less than -30%) of $Q_h^{P1}$ and $Q_h^{P2}$, respectively, which are mostly found in some parts of the Northwest Rivers, and the North China Plain. $Q_h^{P1}$ and $Q_h^{P2}$ are negative for more than 90% of the river segments and range from -5% to 0 in more than 60%

of the river segments of China. The magnitudes of the basin-averaged $Q_h^{P1}$ and $Q_h^{P2}$ are larger than 10% in northern basins except for the Songhua River (Figure 4c). The magnitudes of $Q_h^{P2}$ are larger than $Q_h^{P1}$ for all basins, especially in the Yellow River. The Northwest Rivers show the largest negative values of $Q_h^{P1}$ and $Q_h^{P2}$ (-31.6% and -33.5%, respectively), which is followed by the Hai River (-24% and -25%), the Yellow River (-17% and -21%) and the Huai River (-17% and -19%). DHI induced slight decreases in MAF (-3.4% to -0.3%) in the southern basins. Overall, DHI altered MAF by -4.4% and -5.6% in

China during the sub-periods P1 and P2, respectively.

**3.4 MAF changes induced by DHI change and climate change between the two sub-periods**

The MAF changes induced by DHI change and climate change between the two sub-periods are shown in Figure 5. In general, total MAF changes ($\Delta Q_a$, Figure 5a) are larger in northern basins except the Songhua River than in southern basins. Compared to the first sub-period, in the second sub-period MAF increased by more than 30% in many river segments of the

Northwest Rivers and increased by more than 5% in large parts of the Huai River. MAF increases are also found in considerable areas of southern basins such as the Yangtze River and the Southwest Rivers. MAF decreases are found in most river segments in the Yellow River, the Hai River, and the Liao River. Significant negative values of $\Delta Q_a$ (less than -20%) are found in some river segments in the upper reaches of the Southwest Rivers and some parts of the Northwest Rivers. The total MAF decreased by more than 10% ($\Delta Q_a < -10\%$) and increased by more than 10% ($\Delta Q_a > 10\%$) in about 24% and 17%

of river segments of China, respectively.

MAF changes induced by climate change between the two sub-periods ($\Delta Q_c$, Figure 5b) have very similar spatial patterns as $\Delta Q_a$ (Figure 5a). It indicates that climate change dominates MAF changes during the two sub-periods. The magnitudes of $\Delta Q_c$ are relatively smaller than those of $\Delta Q_a$ in the Hai River and the Yellow River but are larger in the north-western parts of the Northwest Rivers. MAF changes induced by DHI change ($\Delta Q_h$, Figure 5c) are generally large and negative in northern

basins. Larger than 10% decrease in MAF induced by DHI is found in some segments of the Northwest Rivers and the lower reaches of the Huai River, the Hai River, and the Liao River. Positive values of $\Delta Q_h$ are small and are mostly found in



southern river segments. Climate change dominated MAF changes in most river segments (88%) of China (Figure 5d). Only 12% of river segments show MAF changes that are mainly caused by DHI change, which are mostly in the northern basins.

MAF changes induced by climate change ($\Delta Q_c$) versus those induced by DHI change ($\Delta Q_h$) for all river segments are shown in Figure 6a. Note that very few river segments with values of $\Delta Q_c$ (0.9% of total river segments) and $\Delta Q_h$ (0.4%) beyond [-100, 100] are not shown in the figure. Magnitudes of $\Delta Q_c$ are much larger than those of $\Delta Q_h$. The latter ranges -5% to 5% in most (~ 81%) river segments. $\Delta Q_h$ is less than -10% in only about 7% of river segments of China, while even fewer (~ 3%) segments show $\Delta Q_h$ larger than 5%. The values of $\Delta Q_c$ range from -10% to 10% in more than half of river segments and range from -20% to 20% in nearly 80% of river segments (see Table S2 for related numbers). $\Delta Q_h$ is negative in 70% of river segments, while negative values of $\Delta Q_c$ are found in more than half of the river segments of China (see the percentage numbers in Figure 6a and Table S2).

The total MAF spatially-averaged over China decreased by only 1% from the first sub-period to second sub-period (Figure 6b; also see Table S3-S5 for more details of spatially aggregated ensemble members and medians of $\Delta Q_a$, $\Delta Q_c$, and $\Delta Q_h$ in basins). At the basin scale, the magnitudes of MAF changes are usually very small (less than 2%) in southern basins and are relatively large in northern basins (5% to 13%). $\Delta Q_a$ in the Hai River shows the largest decrease of 13%, which is followed by nearly 10% decrease in the Yellow River and a 7% decrease in the Liao River. Increases of total MAF are found in the Northwest Rivers (10%), and the Huai River (1.8%) and the Pearl River (1.3%) and the Southwest Rivers (1.2%), which are consistent with the spatial patterns shown in Figure 5a. DHI change causes decreases in MAF (negative $\Delta Q_h$) in all the basins, resulting in a larger decrease or a smaller increase in $\Delta Q_a$ compared to $\Delta Q_c$. The largest negative values of $\Delta Q_h$ are found in the Northwest Rivers (-8%), the Huai River (-5.4%), and the Hai River (-4.6%, see Table S3-S5). $\Delta Q_h$ is about -2.6% for the Liao River and the Yellow River. $\Delta Q_h$ is only about -0.7% to -0.07% in southern basins. The increase of MAF induced by climate change ($\Delta Q_c$) are the largest in the Northwest Rivers (18%), followed by the Huai River (6%) and the Pearl River (1%), and climate change caused nearly 9% decrease in MAF in the Hai River and the Yellow River.

### 3.5 Water withdrawal and its changes between the two sub-periods

For both sub-periods, the estimates of long-term mean annual water withdrawals are large (more than $100\times10^6$ m$^3$ per year) in many areas of the Huai River, the Hai River and the Yellow River (Figure 7a). Large water withdrawals are also found in some lower reaches of the Yangtze River. In these regions, mean annual water withdrawals are usually larger in the lower reaches compared to the upper reaches, and significantly increased from 1971-1990 to 1991-2010. The relative changes in water withdrawals between the two sub-periods show distinct spatial patterns from northern to southern basins, and generally increased at all river segments of China (Figure 7c). The spatial patterns of changes in water withdrawals resemble those of $\Delta Q_h$, with large values in the Huai River, the Hai River, and the Yellow River, but are relatively smaller in the Northwest Rivers. Similar analysis is performed for changes in total runoff to examine its linkage with streamflow changes. The spatial



patterns of changes in total runoff induced by DHI change between the two sub-periods (Figure 7d) are different from that of $\Delta Q_h$ (Figure 5c). Total runoff changes are positive in most areas of China due to increasing irrigation water (from both local and external sources) which partly becomes return flow, especially in the Northwest Rivers. Large changes are also found in upper and middle reaches in the Yellow River, the Liao River and the Hai River. The change magnitudes are less than the those induced by climate change (not shown here), which is similar as Figure 5d. This indicates that the runoff changes are less linked to streamflow changes in the study period.

## 4 Discussion

The simulated streamflow in China from the ISIMIP2a VARSOC experiment (i.e., simulations with consideration of DHI) is validated against observed streamflow from 44 hydrological stations. While the multimodel ensemble medians match well with observations, the evaluation indicates that the individual simulations of streamflow are subject to considerable uncertainties among models which are especially pronounced in northern basins as indicated by the ratio of interquartile range to median. The simulations of water withdrawals show large deviations from the reported data in many basins, which partly affects the performance of GHMs in streamflow simulations. It should be noted that the over/underestimation of streamflow at these stations do not necessarily indicates the performance of GHMs in the whole basins because of limited stations used in this study.

Simulated annual streamflow has been increasingly affected by human impact, which is more significant in northern basins. Using the multimodel ensemble medians of streamflow, we quantify the DHI on the long-term MAF during two sub-periods 1971-1990 and 1991-2010, and the long-term MAF changes induced by changes in DHI and climate between the two sub-periods. DHI often results in decreased streamflow in China, particularly in northern rivers, through water withdrawals, while results in increased runoff due to return flow from irrigation. Potential implications of the distinct spatial patterns of DHI and its change on streamflow and the associated uncertainties in current assessment are discussed as follows.

### 4.1 DHI considerably altered streamflow in northern basins

DHI causes MAF decreases in both of the sub-periods. At the basin level, DHI resulted in decreases by one-fifth to one-third of the long-term MAF based on $Q_n$ in northern basins and slightly altered MAF in southern basins of China. The spatial patterns of the MAF altered by DHI ($Q_h$) are generally in accordance to those reported by previous studies (Liu and Du, 2017) and are very close to those of irrigated areas of China (see Figure S1). The expansion of agriculture and enhanced irrigation and food demands should be the main reason for the large DHI on streamflow in northern basins (Liu et al. 2015; also see Figure S1c), where agricultural water use accounts for about 70%-90% of total water use as reported by China Water Resources Bulletin from 1997 to 2010. Water withdrawal for irrigation is less due to the large streamflow and relatively





wetter conditions in southern basins. Limited water resources can further amplify the effects of damming on river segments in northern basins (Yang and Lu, 2014) despite the fewer reservoirs therein compared to southern basins (see Figure S1a).

### 4.2 Hydrological effects from DHI change are limited compared with climate change

Though MAF changes between the two sub-periods are relatively small, especially in southern basins, the respective contributions of climate change and DHI change are still distinguishable. In general, streamflow changes are dominated by climate change between the two sub-periods in most river segments of China. Similar results have been reported by a recent study in the United States and Canada (Ficklin et al., 2018). The small portion (12%) of river segments where DHI change outweighs climate change impact on MAF changes are mostly in northern China. The small magnitudes of MAF changes induced by DHI change between the two sub-periods may be partly due to that DHI change is not significant in most areas of China in the VARSOC experiment. Although the irrigated areas in both the northern and southern basins increased by about 20% in the second sub-period (see Figure S1c), the changes between the two sub-periods are small (less than 5%) in many areas except in the Huai River and the Hai River (see Figure S1b). Furthermore, there are only a few reservoir data from the GRanD database after the year 2000, and most reservoirs in China were built in the first sub-period (see Figure S1d). The reservoirs lacking construction years were set to be built (and operated) at the beginning of the experiment in the model runs.

It is noted that the absolute MAF changes between the two sub-periods are large in main streams in both southern and northern basins (see Figure S3); and the significant MAF changes induced by DHI change in the Yangtze River are associated with the large reservoir regulations, e.g., the Three Gorges Reservoir (Wang et al., 2013a).

### 4.3 Water withdrawals are identified as the major DHI to streamflow

Overall, the spatial patterns of water withdrawal changes (Figure 7c) are similar to MAF changes induced by DHI change ($\Delta Q_h$, Figure 5c) between the two sub-periods. Though water use partly infiltrates into land surface and eventually increases local runoff (see Figure 7d), water withdrawals should be the major DHI that contributes to decreases in streamflow in most river segments in China. For example, the significant decreases in MAF are supposed to be largely related to water withdrawals in the Northwest Rivers where streamflow is low and only one reservoir was included in the VARSOC simulations. The water withdrawal changes in Northwest Rivers are relatively small compared to other northern basins, but they still have significant implications because of the limited water resources. As mentioned above, water withdrawal for agricultural irrigation accounts for the largest proportion of human water use in China, most of which evaporates into the atmosphere finally through both crop and soil because of the low irrigation efficiencies (Zhu et al., 2013), which might be the main source depleting the streamflow and local water resources. Though the return flow might increase runoff over most river segments of China (Figure 7d), it seems to be only a small proportion of the water withdrawals and does not offset the decreases in streamflow. Unlike water withdrawals, the effects of reservoir regulation on annual streamflow are mixed in



current GHMs as reservoir regulation generally reduces streamflow in flood (and growing) seasons while streamflow increases in dry seasons (Masaki et al., 2017).

### 4.4 Increasing DHI may impair the adaptive capacity of freshwater system

Though the effects of DHI change on streamflow are smaller compared to those of climate change in China (see section 4.2), the DHI-induced streamflow changes significantly increased particularly in the northern basins over the 1971-2010 period (Figure 3). The northern basins have relatively poor water endowments and have been identified as regions that are highly sensitive to climate change (Piao et al., 2010). The relatively high DHI further increase the pressure and threats to water management and adaptation to future climate change in these regions. For example, frequent zero flow was observed in some reaches of the Yellow River due to climate change and human water use in the 1990s (Tang et al., 2013). Most northern regions suffered severe water scarcity during the past decades (Liu et al., 2017a), and the water resources have been increasingly insufficient for human water needs in many areas of northern basins (Liu and Xia, 2004). The unregulated pumping of non-renewable groundwater has resulted in significant depletion and far-reaching effects on both hydrological cycle and human society in these regions (Feng et al., 2013). The DHI change over time further enlarges associated streamflow changes in these basins (see Figure 4c and Figure S4). The situation could be worse if no adaptation is taken to act under future climate change (Piao et al., 2010; Liu et al., 2015). Thus, in view of the considerable DHI in these regions, there is an urgent need for a structural transformation of the economy towards reducing water use and a sustainable development.

### 4.5 Uncertainties in the quantitative assessment

The major uncertainty in this quantitative assessment usually originates from input forcings (Müller Schmied et al., 2014) and inter-model differences such as human impact parameterizations (Liu et al., 2017b). That is, the uncertainties in streamflow simulations would propagate to the assessment. For example, there are very few meteorological observations in the Northwest Rivers, possibly leading to considerable uncertainties in the meteorological forcings used to drive GHMs. Furthermore, the GHMs cannot fully reflect sectoral water withdrawals (Huang et al., 2018; also see Figure 2) because of lacking data on water abstractions for human use from surface and groundwater sources (Liu et al., 2017b). The multimodel ensemble medians seem to be in line with observations averaged across the stations in China, but large discrepancies are found in some basins (Figure S2). This indicates a large space for the GHMs to improve streamflow simulations in China. It should be noted that we have relocated some stations on the map to reconcile the catchment areas of the stations and the corresponding grid cells on the DDM30 river network. However, catchment areas still are inconsistent between some stations and their corresponding grid cells, especially for the stations not on the main stream. This may be partly responsible for the deviation between simulated and observed streamflow. More hydrological observations (from large catchment areas) are necessary to perform a comprehensive evaluation of streamflow simulations.





In addition to the uncertainties in multimodel simulations of streamflow, the quantitative assessment depends on the selection of comparison periods (see Table S7). To examine the possible effects of the selection of sub-periods, we perform similar assessments for different sub-periods, i.e., MAF changes of three decades of 1981-1990, 1991-2000, and 2001-2010 compared to the first decade (1971-1980). The assessments show similar patterns of MAF changes as in Figure 5, with larger

relative changes in most northern basins (see Figure S4 for the analysis at basin scale). Effects of climate change on streamflow vary over different sub-periods. In contrast, DHI change usually resulted in MAF decrease across all basins and its impact slightly increases over time (see Table S6 for corresponding numbers), especially in the northern basins such as the Yellow River, the Northwest Rivers, the Liao River, and the Hai River. In the Yellow River, MAF changes induced by DHI change outweigh that induced by climate change in the 2001-2010 period. Human activities may be weaker in China

before the year 1971, and the DHI change could be larger if compared to earlier periods (e.g., Müller Schmied et al., 2016). This assessment suggests that the magnitudes of the impacts of both climate change and DHI change on streamflow are associated with specific sub-periods, however, DHI change decreased streamflow in almost all basins in the study period.

### 4.6 Comparison with previous studies

Both this study and previous ones (see Table S7) show that DHI (change) almost always contributes to decreases in

streamflow in China, but the DHI contributions are much more significant in previous assessments compared to this one. Previous studies have shown that DHI contributed to decrease in streamflow by 20% to 80% across catchments in the Hai River, Yellow River and Huai River (see Table S7, it should be noted that the proportions in the table were calculated as $100 \times \Delta Q_h / \Delta Q_a$). In four cases the DHI contributions are larger than those of climate change impact and in most cases DHI contribute more than 40% in these studies (see Table S7 for the results from previous studies), while DHI contributions are

mostly smaller than climate change in this assessment (Figure 6a). There are several reasons for the large differences between this assessment and previous ones which make their results not comparable directly, such as different methods and data, sub-periods, and study areas (see Figure S7 for details). The previous assessments were usually performed in small catchments where experience evident human activities and usually chose comparison periods by using statistical approaches (e.g., abrupt-changing point detection for a time series).

In contrast with previous studies, the multimodel simulations facilitate the attribution of DHI (change) on streamflow in a consistent manner that is largely free of the uncertainty of data from different systems (i.e., modeling and observation). They also allow profiling the uncertainties among models and input forcings, which is difficult for a single model assessment. Considering the complexity of DHI on streamflow and the ability of current hydrological models in reproducing historical hydrological changes, multimodel simulations and different attribution approaches are well worth obtaining more robust

assessments (Liu et al., 2017b; Yuan et al., 2018).





## 5 Conclusions

A quantitative assessment of the contributions of DHI (direct human impact) and climate change impact on streamflow changes is performed in the ten major river basins in China during the 1971-2010 period. The ISIMIP2a multimodel simulations are evaluated against hydrological observations in China and are used for the assessment. The results show that

DHI caused decreases of one-fifth to one-third of the long-term MAF in the sub-periods of 1971-1990 and 1991-2010 in most northern basins. MAF changes between the two sub-periods are small in southern basins but are relatively large in northern basins where MAF decrease by 10% or more. It is found that DHI change between the two sub-periods resulted in MAF decreases in 70% of the river segments. However, total MAF changes are dominated by climate change in 88% of the river segments of China. The respective contributions of climate and DHI changes to streamflow changes are more

pronounced in northern basins. The relative contribution of DHI change shows significant regional difference with relatively larger values in northern basins (-3% to -8% of MAF) and smaller ones in southern basins (-0.7% to -0.07%). The contribution of climate change to streamflow changes varies between basins, ranging from -9% to 18% of MAF in northern basins and from -1.6% to 1.3% in southern basins. The same analyses for different sub-periods, i.e., the 1980s, 1990s, and 2000s compared with the 1970s, show similar spatial patterns of the contribution of DHI change. It indicates that human

intervention is high in northern basins with an increasing trend over time, which likely impairs the adaptive capacity of freshwater system under future climate change. This assessment also shows that water withdrawals are the major factor that directly affects streamflow in China. It should be noted that this assessment is subject to uncertainties arising from the uncertainties in multimodel simulations and the choice of study periods. Nevertheless, it can serve as a reference, in a socio-hydrological perspective, for the attribution of changes in streamflow at large scales under a changing environment. We

highlight the importance of reducing DHI on streamflow for a sustainable development in northern basins of China and expect the assessment to favor China's strategy on adaptation to future climate change.

*Data availability*. All model data used in this study can be accessed by the public following the instructions on the website of the Inter-Sectoral Impact Model Intercomparison Project (www.isimip.org).

*Author contribution*. XL, QT, WL, HY designed the research; XL, MF, YM, HMS, SO, YP, YS, YW prepared the model

data; XL performed the analyses and wrote the draft, and all authors wrote the manuscript.

*Competing interests*. The authors declare that they have no conflict of interest.

## Acknowledgments

We thank the Inter-Sectoral Impact Model Intercomparison Project coordinating team for providing the simulated data. This research is supported by the National Natural Science Foundation of China (41730645, 41425002, 41790424, and 41877164),

the Key Research Program of the Chinese Academy of Sciences (KGFZD-135-17-009-3, ZDRW-ZS-2017-4), and the





International Partnership Program of Chinese Academy of Sciences (131A11KYSB20170113). W.L. acknowledges the support received from the Early Postdoctoral Mobility Fellowship awarded by the Swiss National Science Foundation (P2EZP2_175096). Y.P. acknowledges the support from the Asian Studies Center at Michigan State University.

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





**Figure 1. Multimodel medians of mean annual streamflow (MAF) in China from the VARSOC experiment. MAF medians are computed across 18 GHM-GMF combinations over the 1971-2000 period. The ensemble spread is represented by the ratio of interquartile range (IQR, 75th percentile minus 25th percentile) to the ensemble median of MAF (Median). The hydrological stations used in this study are identified by red circles. The inner plot shows the comparison of the simulated seasonal streamflow (each GHM has three lines for the three GMFs) from the VARSOC experiment against the observations averaged for all the hydrological stations shown on the map over the period 1971-2000. H08: H08, DBH: DBH, LPJ: LPJmL, PCR: PCR-GLOBWB, WAT: WaterGAP2, MAT: MATSIRO, MME: multimodel ensemble median, OBS: observation. The Tibetan Plateau region is masked by removing the grid cells with an altitude higher than 4000 meters, the same hereafter. The ten major basins in China are labeled and are indicated with grey lines. The southern basins include Yangtze River (YZ), Southwest Rivers (SW), Southeast Rivers (SE), and Pearl River (PR), the northern basins include Songhua River (SH), Liao River (LR), Northwest Rivers (NW), Hai River (HA), Yellow River (YR), and Huai River (HU).**



**Figure 2. Reported and simulated water withdrawals in the 10 basins of China. ISIMIP2a indicates the simulated water withdrawals from the ISIMIP2a VARSOC experiment (see Table S1 for details) during 1971-2010; MWR indicates the water withdrawals reported by the Ministry of Water Resources (MWR) of China for the years of 1980, 1985, 1990, 1995, and 1997-2010.**
5 **Δ indicates the difference between simulations and reported data. Shaded areas denote the IQR of ISIMIP2a simulations. The basin names labeled in each panel are corresponding to the basins in Figure 1.**



**Figure 3. Spatially-averaged annual streamflows (m³ s⁻¹) from NOSOC and VARSOC experiments and their differences (%) during the 1971-2010 period. (a) Average of ensemble medians of annual streamflow from NOSOC ($Q_n$) and VARSOC ($Q_v$) for China, (b) for the northern basins, and (c) for the southern basins. The northern and southern basins are described in Figure 1. The dashed lines denote the linear trend of the relative differences.**






**Figure 4. Long-term MAF altered by DHI. Ensemble medians of long-term MAF altered by DHI in (a) the sub-period 1971-1990 ($Q_h^{P1}$) and (b) the sub-period 1991-2010 ($Q_h^{P2}$), and (c) ensemble medians and ranges of averaged long-term MAF altered by DHI for each basin and China (denoted by CN). In plot (c), the range indicates the 25th and 75th values, and the numbers indicate the median values from all ensemble members.**





**Figure 5. Relative changes (%) in long-term MAF over China between the two sub-periods (1971-1990 and 1991-2010). (a): Total MAF changes ($\Delta Q_a$); (b): MAF changes induced by climate change ($\Delta Q_c$); (c): MAF changes induced by DHI changes ($\Delta Q_h$); (d): the difference between the magnitudes of $\Delta Q_c$ and $\Delta Q_h$.**





**Figure 6. Relative MAF changes for river segments and basins. (a): ensemble medians of MAF changes induced by climate change ($\Delta Q_c$) versus those induced by DHI change ($\Delta Q_h$) for river segments of China; data points in (a) denote the values for individual river segments; the right histogram and the top histogram show the distributions of $\Delta Q_c$ and $\Delta Q_h$, respectively; the numbers are the proportions of data points in each quadrant. (b): spatially aggregated ensemble medians of total MAF changes ($\Delta Q_a$), MAF changes induced by climate change ($\Delta Q_c$), and MAF changes induced by DHI change ($\Delta Q_h$) for individual basins and China; the error bars indicate the IQR in each basin.**





**Figure 7. Changes in water withdrawals and total runoff between the two sub-periods. (a): ensemble medians of mean annual water withdrawals over the1971-1990 period; (b): ensemble medians of mean annual water withdrawals over 1991-2010 period; (c) ensemble medians of the changes in mean annual water withdrawals; (d): ensemble medians of the changes in mean annual total runoff.**