# Peer review of "Multimodel assessments of human and climate impacts on mean annual streamflow in China"

_Hydrology and Earth System Sciences, 2018_

## Referee Comment (RC1) · Anonymous Referee #1 · 3 Nov 2018

This study quantitative assessed the human impact and climate change impact on streamflow in continental China. The simulations streamflow was used from six global hydrological models driven by three meteorological forcings. The research is very interesting and significative. However, there are a few issues that the authors need to address before the manuscript can be accepted. I recommend most of the issues I raise below just need clarification or justification. 1. The simulated results need to be verified further with observed streamflow, maybe, QQPLOT, NSE etc. method can be used. 2. The simulated results are very bad in some basins, such as NW, SW, HA. These simulated streamflow need be post-processed, and then be used to analyzed the impact of human and climate change. 3. The authors need add some explanation of ISIMIP2a about how to simulate water withdrawals.

---

## Referee Comment (RC2) · Anonymous Referee #2 · 27 Nov 2018

The paper describes a multimodel assessment of the relative impacts of human activities and climate on mean annual streamflow over the past 4 decades in China. This study shows that unlike previous assessments, the climate impact signal is much more pronounced than the human impact signal in 88% of river segments in China. The study also quantifies the impact of humans across basins and discusses regional differences. In general the paper is publishable after some moderate revisions.

- The use of the term 'climate change' in the title and throughout the manuscript is somewhat confusing and misleading because it gives the impression that the paper will be forward looking in time and over the coming several decades (e.g., 2050, 2100). A more appropriate term is 'climate impacts'
- The 3rd paragraph of the introduction makes the argument that "This is the first study to perform such a quantitative assessment for all rivers of China with comparable modeling experiments." Being aware of the ISIMIP publications (https://www.isimip.org/outcomes/publications/) in this space with global assessments including many of the authors on this paper, I find this argument to be an exaggeration. I think the last sentence of that paragraph is a key novelty of this work, and as such linking back to the content of the second paragraph in the introduction to make the case would be my suggestion. I do agree that focusing on China is somewhat unique about this study. So one suggestion is to tweak the noted sentence as follow "This is the first study to focus on performing such a quantitative assessment for all rivers of China with comparable modeling experiments."
  - o Schewe et al.: Multimodel assessment of water scarcity under climate change. PNAS, 2014.
  - o Haddeland et al.: Global water resources affected by human interventions and climate change, P. Natl. Acad. Sci. USA, 111, 3251–3256, https://doi.org/10.1073/pnas.1222475110, 2014.
  - o Veldkamp et al.: Water scarcity hotspots travel downstream due to human interventions in the 20th and 21st century, Nature Commun., 8, 15697, https://doi.org/10.1038/ncomms15697, 2017.
  - o Wada et al.: Human–water interface in hydrological modelling: current status and future directions, Hydrol. Earth Syst. Sci., 21, 4169-4193.
- P5, L2: I would suggest omitting 'preliminary'
- P7, L14-17: Showing the individual models in figure S2 makes the figure too busy to read. Why not use the same format as in figure 2 by showing a band around the median. Also, it would be useful to show the same type of figure as figure 2 but for streamflow.
- P7, L18-24: I realize that given the large departures in water withdrawal estimates, matching streamflow guage observations might be a challenge, unless the authors believe that simulated water withdrawals might be equally or even more reliable than the statistically collected data, which have their own challenges.
- P7, L18-24: Are water withdrawals taken from surface water sources or also groundwater sources? What about return flows? Also, I am assuming that glacier melting, which contributes to streamflow, is simulated in these models, but that region is not included in the analysis. I realize that some of these were mentioned in the results, but incorporating some of these

details briefly when discussing the method or the results from the evaluation exercise would suffice.

- P10, L23-29: how does the model specify how much water is taken from surface water vs groundwater sources? Are the small pockets of increased MAF due to human impacts (Fig 5c) attributed to technological change (e.g., irrigation efficiency), or return flow from groundwater pumping, or something else?
- P11, L7: I would suggest omitting the sentence about the US and Canada. It breaks the flow of the paragraph which is talking specifically about China.
- P13, L14-30: To me this, this is a key contribution of this study. Yes, I agree that the results are not necessarily comparable in term magnitudes due to the highlighted reasons by the authors. But a missing discussion point is to why they fundamentally differ in their findings. I don't agree that either one of these two approaches (small scale using statistical approaches vs large scale modeling similar to this study) is necessarily superior. Each approach has its own pros and cons. So articulating why this approach differs from earlier findings is critical.

---

## Author Comment (AC1) · 2 Jan 2019

We thank the reviewers for their valuable comments. We have replied the comments one by one below, and revised the manuscript accordingly. The replies are highlighted in blue color and the modified texts (in the revised manuscript) are shown italic.

Reviewer #1

This study quantitative assessed the human impact and climate change impact on streamflow in continental China. The simulations streamflow was used from six global hydrological models driven by three meteorological forcings. The research is very interesting and significative. However, there are a few issues that the authors need to address before the manuscript can be accepted. I recommend most of the issues I raise below just need clarification or justification. Reply: Thanks for the positive comment. We have replied the comments below and revised manuscript accordingly.

1. The simulated results need to be verified further with observed streamflow, maybe, QQPLOT, NSE etc. method can be used.
Reply: We have calculated NSE for the 44 stations, and added a sentence describing the result with a table (new added Table S2) in the revised supplementary information.

Revision in the manuscript (Subsection 3.1, the second paragraph (new added)):
*The Nash-Sutcliffe coefficients calculated for the multimodel median and observed monthly streamflow at each station (see Table S2) show that the multimodel medians have better performance in the southern basins.*

*Table S2. The Nash-Sutcliffe coefficients (NSE) for the simulated monthly streamflow from VARSOC experiment and observed monthly streamflow ($m^3 s^{-1}$) at the 44 stations over the 1971-2000 period. The observed mean annual streamflow (MAF, $m^3 s^{-1}$) averaged over the period is also shown for each station.*

| Number | Station Name | MAF | NSE | River name | Number | Station Name | MAF | NSE | River name |
|--------|--------------|-----|-----|------------|--------|--------------|-----|-----|------------|
| 1 | Guchengzi | 151.26 | -0.27 | Songhua River | 23 | Xixian | 117.87 | 0.31 | Huai River |
| 2 | Fuyu | 449.68 | 0.53 | Songhua River | 24 | Fuyang | 117.74 | 0.63 | Huai River |
| 3 | Tonghe | 1444.43 | 0.81 | Songhua River | 25 | Lutaizi | 639 | 0.80 | Huai River |
| 4 | Kuerbin | 26.94 | 0.004 | Songhua River | 26 | Bengbu | 800.63 | 0.81 | Huai River |
| 5 | Chaoyang | 18 | -0.88 | Liao River | 27 | Shishang | 1968.28 | 0.93 | Yangtze River |
| 6 | Chifeng | 7.76 | -0.25 | Liao River | 28 | Changyang | 431.02 | 0.75 | Yangtze River |
| 7 | Tieling | 84.36 | <-1.0 | Liao River | 29 | Pingshan | 4546.38 | 0.77 | Yangtze River |
| 8 | Liaozhong | 101.43 | 0.50 | Liao River | 30 | Sinan | 910.98 | 0.79 | Yangtze River |
| 9 | Changmapu | 29.3 | -0.37 | Northwest Rivers | 31 | Cuntan | 10747.92 | 0.68 | Yangtze River |
| 10 | Yingluoxia | 51.06 | -0.26 | Northwest Rivers | 32 | Datong | 28460.19 | 0.78 | Yangtze River |
| 11 | Zhamashenke | 22.7 | 0.09 | Northwest Rivers | 33 | Quzhou | 207.66 | 0.80 | Southeast Rivers |
| 12 | Sandaohezi | 16.45 | <-1.0 | Hai River | 34 | Zhuji | 40.24 | 0.58 | Southeast Rivers |
| 13 | Panjiakou | 60.87 | 0.01 | Hai River | 35 | Zhuqi | 1721.14 | 0.91 | Southeast Rivers |
| 14 | Luanxian | 96.09 | 0.71 | Hai River | 36 | Yangkou | 442.85 | 0.72 | Southeast Rivers |
| 15 | Xiapu | 4.58 | <-1.0 | Hai River | 37 | Daojieba | 1746.97 | 0.12 | Southwest Rivers |
| 16 | Huangbizhuang | 32.14 | -0.07 | Hai River | 38 | Gulaohe | 96.63 | 0.22 | Southwest Rivers |
| 17 | Cetian | 4.78 | -0.01 | Hai River | 39 | Manhao | 310.84 | 0.82 | Southwest Rivers |
| 18 | Lanzhou | 976.8 | 0.53 | Yellow River | 40 | Jiangbianjie | 194.96 | 0.68 | Pearl River |
| 19 | Shizuishan | 867.25 | 0.45 | Yellow River | 41 | Duanzhan | 2005.11 | 0.88 | Pearl River |
| 20 | Longmen | 803.67 | -0.47 | Yellow River | 42 | Xiayan | 449.63 | 0.82 | Pearl River |
| 21 | Huayuankou | 1103.51 | 0.09 | Yellow River | 43 | Wuxuan | 4130.25 | 0.81 | Pearl River |
| 22 | Xianyang | 107.26 | 0.63 | Yellow River | 44 | Boluo | 782.04 | 0.80 | Pearl River |

2. The simulated results are very bad in some basins, such as NW, SW, HA. These simulated streamflow need be post-processed, and then be used to analyzed the impact of human and climate change.

Reply: We recognized the poor performance of the simulations, especially in the northern basins (see above Table S2). A post-processing on the simulations could reduce the deviation in simulated streamflow from observations and narrow the spread across models (e.g., Yin et al., 2017). However, the post-processing can be affected by the distribution and the number of the stations. In this study, though we have collected hydrological observations from 44 stations in China, they may be not representative enough for all basins. For example, there are only three stations in the Northwest Rivers (NW) which cover small areas. More stations are also needed for the basins like the Southwest Rivers where the streamflow changes greatly from upstream to downstream. Therefore, we think post-processing is not appropriate to and not necessarily improve the streamflow simulations in this study based on multimodel simulations. Furthermore, we tend to focus on the multimodel uncertainty in the model results in the evaluation section. We have added a caution for the limited representative of the observations in the evaluation result to remind readers to treat it carefully.

Revision in the manuscript (Subsection 3.1, the second paragraph (new added)):
It should be noted that the stations are located at different reaches of individual basins. Thus, the station-averaged estimates are largely dominated by those with large streamflow (e.g., at the lower reaches). *Additionally, the coverage of stations used is relatively small (due to data availability), especially in hydrologically variable regions like in the Northwest Rivers, leading to not necessarily representative evaluation of the performance of the GHMs in the whole basin.*

3. The authors need add some explanation of ISIMIP2a about how to simulate water withdrawals.

Reply: We have added some description for the simulated water withdrawals in section 2.1 Simulation data.

Revision in the manuscript (Subsection 2.1, the second paragraph (new added)):
*Human impact considered in the VARSOC experiment (see the maps in Figure S1 and Table S1 for more details) includes the time-varying areas for both irrigated and rainfed cropland (Fader et al., 2010; Portmann et al., 2010) and reservoirs (dams) from the Global Reservoir and Dam (GRanD) Database (Lehner et al., 2011) including their commissioning year (see Figure S1 and Table S1 for more detail). Reservoir regulation was considered in the VARSOC experiment, which often reduces high streamflow in high-flow seasons and increases streamflow in dry seasons (Masaki, et al., 2017). Inter-basin water transfer was not considered in any of the model runs. The simulations of water withdrawals are different between the GHMs with respect to water use requirements and water withdrawal sources which are shown in Table S1. The sources of water withdrawals, depending on models, may include river channel, reservoirs, groundwater and lakes, and their fractions can be determined from reported statistics (e.g., Siebert et al., 2010) or estimated in models (Wada et al., 2014). In addition to the irrigation water requirement which is usually estimated by coupling crop models, most*

*GHMs considered the requirements for domestic and industrial water use which were prescribed in H08 (Hanasaki et al., 2008), LPJmL and MATSIRO (Pokhrel et al., 2015) or were estimated according to the population, socioeconomic and technological development in PCR-GLOBWB (Wada et al., 2014) and the population, thermal electricity production, gross added value, and technological change in WaterGAP (Flörke et al., 2013). Water use requirement for livestock was also prescribed in the LPJmL model, and estimated according to livestock densities in PCR-GLOBWB and WaterGAP2.*

**Reference:**

Fader, M., S. Rost, C. Müller, A. Bondeau, and D. Gerten (2010), Virtual water content of temperate cereals and maize: Present and potential future patterns, J. Hydrol., 384(3–4), 218-231, doi: 10.1016/j.jhydrol.2009.12.011.

Flörke, M., E. Kynast, I. Bärlund, S. Eisner, F. Wimmer, and J. Alcamo (2013), Domestic and industrial water uses of the past 60 years as a mirror of socio-economic development: A global simulation study, Global Environ. Change, 23(1), 144-156, doi: 10.1016/j.gloenvcha.2012.10.018.

Hanasaki, N., S. Kanae, T. Oki, K. Masuda, K. Motoya, N. Shirakawa, Y. Shen, and K. Tanaka (2008), An integrated model for the assessment of global water resources – Part 1: Model description and input meteorological forcing, Hydrol. Earth Syst. Sci., 12(4), 1007-1025, doi: 10.5194/hess-12-1007-2008.

Lehner, B., et al. (2011), High-resolution mapping of the world's reservoirs and dams for sustainable river-flow management, Frontiers in Ecology and the Environment, 9(9), 494-502, doi: 10.1890/100125.

Masaki, Y., N. Hanasaki, H. Biemans, H. Müller Schmied, Q. Tang, Y. Wada, S. N. Gosling, K. Takahashi, and Y. Hijioka (2017), Intercomparison of global river discharge simulations focusing on dam operation—multiple models analysis in two case-study river basins, Missouri–Mississippi and Green–Colorado, Environ. Res. Lett., 12(5), 055002.

Pokhrel, Y. N., S. Koirala, P. J. F. Yeh, N. Hanasaki, L. Longuevergne, S. Kanae, and T. Oki (2015), Incorporation of groundwater pumping in a global Land Surface Model with the representation of human impacts, Water Resour. Res., 51(1), 78-96, doi: 10.1002/2014WR015602.

Portmann, F. T., S. Siebert, and P. Döll (2010), MIRCA2000—Global monthly irrigated and rainfed crop areas around the year 2000: A new high-resolution data set for agricultural and hydrological modeling, Global Biogeochem. Cycles, 24(1), GB1011, doi: 10.1029/2008GB003435.

Siebert, S., J. Burke, J. M. Faures, K. Frenken, J. Hoogeveen, P. Döll, and F. T. Portmann (2010), Groundwater use for irrigation – a global inventory, Hydrol. Earth Syst. Sci., 14(10), 1863-1880, doi: 10.5194/hess-14-1863-2010.

Yin, Y., Q. Tang, X. Liu, and X. Zhang (2017), Water scarcity under various socio-economic pathways and its potential effects on food production in the Yellow River basin, Hydrol. Earth Syst. Sci., 21(2), 791-804, doi: 10.5194/hess-21-791-2017.

Wada, Y., D. Wisser, and M. F. P. Bierkens (2014), Global modeling of withdrawal, allocation and consumptive use of surface water and groundwater resources, Earth Syst. Dynam., 5(1), 15-40, doi: 10.5194/esd-5-15-2014.

---

## Author Comment (AC2) · 2 Jan 2019

We thank the reviewer for the valuable comments. We have replied the comments one by one below, and revised the manuscript accordingly. The replies are highlighted in blue color and the modified texts (in the revised manuscript) are shown italic.

Reviewer #2

The paper describes a multimodel assessment of the relative impacts of human activities and climate on mean annual streamflow over the past 4 decades in China. This study shows that unlike previous assessments, the climate impact signal is much more pronounced than the human impact signal in 88% of river segments in China. The study also quantifies the impact of humans across basins and discusses regional differences. In general the paper is publishable after some moderate revisions.

Reply: Thanks for the positive comments. We have replied the comments and revised the manuscript accordingly.

- The use of the term 'climate change' in the title and throughout the manuscript is somewhat confusing and misleading because it gives the impression that the paper will be forward looking in time and over the coming several decades (e.g., 2050, 2100). A more appropriate term is 'climate impacts'

Reply: We used "climate impact" in the title. We have replaced "climate change" by "climate variability" in most cases, and by "climate impact" in some cases where appropriate. "climate change" is still used when refer to future climate change and results from some specific references.

- The 3rd paragraph of the introduction makes the argument that "This is the first study to perform such a quantitative assessment for all rivers of China with comparable modeling experiments." Being aware of the ISIMIP publications (https://www.isimip.org/outcomes/publications/) in this space with global assessments including many of the authors on this paper, I find this argument to be an exaggeration. I think the last sentence of that paragraph is a key novelty of this work, and as such linking back to the content of the second paragraph in the introduction to make the case would be my suggestion. I do agree that focusing on China is somewhat unique about this study. So one suggestion is to tweak the noted sentence as follow "This is the first study to focus on performing such a quantitative assessment for all rivers of China with comparable modeling experiments."
    o Schewe et al.: Multimodel assessment of water scarcity under climate change. PNAS, 2014.
    o Haddeland et al.: Global water resources affected by human interventions and climate change, P. Natl. Acad. Sci. USA, 111, 3251–3256, https://doi.org/10.1073/pnas.1222475110, 2014.
    o Veldkamp et al.: Water scarcity hotspots travel downstream due to human interventions

in the 20th and 21st century, Nature Commun., 8, 15697, https://doi.org/10.1038/ncomms15697, 2017.

  o Wada et al.: Human–water interface in hydrological modelling: current status and future directions, Hydrol. Earth Syst. Sci., 21, 4169-4193.

Reply: Thanks for the suggestion. We have revised the sentence following the suggestion.

  - P5, L2: I would suggest omitting 'preliminary'

Reply: Changed.

  - P7, L14-17: Showing the individual models in figure S2 makes the figure too busy to read. Why not use the same format as in figure 2 by showing a band around the median. Also, it would be useful to show the same type of figure as figure 2 but for streamflow.

Reply: We have redrawn Figure S2 and added a figure (Figure S3 in the revised manuscript and below) for the simulated and observed annual streamflow following the suggestion.

A brief description has been added.

Revision in the manuscript (Subsection 3.1, the second paragraph (new added)):

*The model spreads in the ensembles of seasonal streamflow and the deviations between observation and simulations are relatively larger in the northern basins than those in the southern basins (see Figure S2 for each basin). Comparison between the simulated and observed annual streamflow (Figure S3) shows similar patterns as the seasonal streamflow with respect the discrepancies between northern and southern basins. The Nash-Sutcliffe coefficient was calculated for the multimodel median and observed monthly streamflow at each station (see Table S2), which shows that the multimodel medians have better performance in the southern basins. This evaluation indicates that the multimodel simulations have relatively poor performance in northern basins and most stations with relatively smaller streamflow (e.g., in dry areas or upper reaches). The large spreads between models underline the necessity of using ensemble medians rather than individual models for the attribution of streamflow changes.*

[Figure]

*Figure S2. Seasonal cycle of streamflow from observations (orange) and multimodel medians (black). The observations are the average values of the hydrological stations, while the simulations are averaged values over the grid cells identified by the location of stations. The gray areas show the 25th and 75th percentiles of the multimodel simulations. Northern basins: Songhua River (SH), Liao River (LR), Northwest Rivers (NW), Hai River (HA), Yellow River (YR), Huai River (HU); Southern basins: Yangtze River (YZ), Southeast Rivers (SE), Southwest Rivers (SW), Pearl River (PR).*

[Figure]

*Figure S3. Simulated (black) and observed (orange) mean annual streamflow at the hydrological stations in each basin. The observations are the average values of the hydrological stations, while the simulations are averaged values over the grid cells identified by the location of stations. The gray areas show the 25th and 75th percentiles of the multimodel simulations.*

- P7, L18-24: I realize that given the large departures in water withdrawal estimates, matching streamflow gauge observations might be a challenge, unless the authors believe that simulated water withdrawals might be equally or even more reliable than the statistically collected data, which have their own challenges.

Reply: We would not state that the simulations of water withdrawals are equally or more reliable than the statistically collected data. In fact, they are largely based on statistically collected data (e.g., Flörke et al. 2013; Hanasaki et al., 2008). We agree that the large spreads in the multimodel simulations of water withdrawals should be one uncertainty source to the streamflow simulations, but the effect is not superimposing (e.g., Fig. 9 in Müller Schmied et al 2014 for one GHM). On the other hand, Veldkamp et al. (2018) showed that inclusion of human impacts such as water withdrawals leads to better model performances. The simulations of water withdrawals remain a challenge, though great efforts have been made by the community. We have added a statement to address this concern at the end of this paragraph in the revised manuscript, but would not further elaborate it since it is not the focus of this study.

Revision in the manuscript (Subsection 3.1, the last paragraph)
*The large deviations in the multimodel simulations of water withdrawals could make the modeling of streamflow more challenging (Döll et al., 2016; Wada et al., 2017).*

- P7, L18-24: Are water withdrawals taken from surface water sources or also groundwater sources? What about return flows? Also, I am assuming that glacier melting, which contributes to streamflow, is simulated in these models, but that region is not included in the analysis. I realize that some of these were mentioned in the results, but incorporating some of these details briefly when discussing the method or the results from the evaluation exercise would suffice.
Reply: The sources of water withdrawals are shown in Table S1, depending on models, which may include river channel, reservoirs, groundwater and lakes. Return flows were considered in different ways for different water uses (e.g., Müller Schmied et al., 2014; Wada et al., 2014; Pokhrel et al., 2015). Glacier melting was not simulated in most GHMs (except PCR-GLOBWB) in this study. We have added the absence of glacier melting in the models as a reason of excluding the Tibet plateau region, and described the sources of water withdrawals in the Method section (please see the reply to the 3rd comment of reviewer #1) in the revised manuscript.

Revision in the manuscript (Subsection 2.1, the first paragraph)
The simulations may have large uncertainties over the Tibetan Plateau because long-term meteorological and streamflow observations are sparse in this region (Zhang et al., 2017) *and the modeling of glacier melting is absence in most of the models.*

- P10, L23-29: how does the model specify how much water is taken from surface water vs groundwater sources? Are the small pockets of increased MAF due to human impacts (Fig 5c) attributed to technological change (e.g., irrigation efficiency), or return flow from groundwater pumping, or something else?
Reply: Generally, groundwater is withdrawn when the water use requirement is not met due to limited accessibility to or insufficient quality of surface water. Groundwater withdrawal was considered in most models (see Table S1), but the pumping rate may vary substantially between

models (Wada et al., 2016). It is difficult to determine the groundwater pumping rate since groundwater storage is usually unknown. The fraction of groundwater for water use is determined from reported statistics data (Siebert et al., 2010, used in WaterGAP) or estimated in the model (e.g., PCR-GLOBWB, see Wada et al., 2014). We have briefly clarified it in the Method section (also see the reply to the 3rd comment of reviewer #1) in the revised manuscript.

The increased MAF should be mainly due to return flow, but we cannot identify it from which source because of lacking related model output currently. Technological development may improve water use efficiency and reduce the amount of withdrawals. However, it may be not the reason for the slight MAF increase induced by DHI change, because water withdrawal increased over the study period (see Figure 2). We have clarified it in the revised manuscript.

Revision in the manuscript (Subsection 3.4, the first paragraph)
Compared to the first sub-period, in the second sub-period MAF increased by more than 30% in many river segments of the Northwest Rivers and increased by more than 5% in large parts of the Huai River, *which may be due to the return flow from water withdrawals.*

- P11, L7: I would suggest omitting the sentence about the US and Canada. It breaks the flow of the paragraph which is talking specifically about China.
Reply: Removed.

- P13, L14-30: To me this, this is a key contribution of this study. Yes, I agree that the results are not necessarily comparable in term magnitudes due to the highlighted reasons by the authors. But a missing discussion point is to why they fundamentally differ in their findings. I don't agree that either one of these two approaches (small scale using statistical approaches vs large scale modeling similar to this study) is necessarily superior. Each approach has its own pros and cons. So articulating why this approach differs from earlier findings is critical.
Reply: We agree that both the methods has its own pros and cons. At the end of the discussion, we have emphasized the importance of using multiple approach to obtain more reliable assessment. In the revised manuscript, we have rewritten the last paragraph of the discussion, wherein we clearly stated the key difference between the method in previous studies and this study.

Revision in the manuscript (Subsection 4.6, the last paragraph)
*One major difference between previous studies (e.g., Li et al., 2007; Bao et al., 2012) and this study is that the former estimates DHI contribution by comparing simulations with observations while we compare two simulation experiments. The former may be subject to uncertainty in comparing the data from two systems (i.e., the model and the real world). In this study, the two simulation experiments favor the estimation of DHI contribution in a consistent manner that is largely free of uncertainty in the data from different systems. The multimodel simulations also allow profiling the uncertainties among models and input forcings, which is difficult for a single*

*model assessment. However, the deficiency of this approach is that DHI is not real. Therefore, the assessment is inevitably influenced by the extent to which the models can reproduce the real DHI.* Considering the complexity of DHI on streamflow and the ability of current hydrological models in reproducing historical hydrological changes, multimodel simulations and different attribution approaches are well worth obtaining more robust assessments (Liu et al., 2017; Yuan et al., 2018).

**Reference:**
Bao, Z., J. Zhang, G. Wang, G. Fu, R. He, X. Yan, J. Jin, Y. Liu, and A. Zhang (2012), Attribution for decreasing streamflow of the Haihe River basin, northern China: Climate variability or human activities?, J. Hydrol., 460-461, 117-129, doi: 10.1016/j.jhydrol.2012.06.054.

Döll, P., H. Douville, A. Güntner, H. Müller Schmied, and Y. Wada (2016), Modelling Freshwater Resources at the Global Scale: Challenges and Prospects, Surv. Geophys., 37(2), 195-221, doi: 10.1007/s10712-015-9343-1.

Flörke, M., E. Kynast, I. Bärlund, S. Eisner, F. Wimmer, and J. Alcamo (2013), Domestic and industrial water uses of the past 60 years as a mirror of socio-economic development: A global simulation study, Global Environ. Change, 23(1), 144-156, doi: 10.1016/j.gloenvcha.2012.10.018.

Hanasaki, N., S. Kanae, T. Oki, K. Masuda, K. Motoya, N. Shirakawa, Y. Shen, and K. Tanaka (2008), An integrated model for the assessment of global water resources – Part 1: Model description and input meteorological forcing, Hydrol. Earth Syst. Sci., 12(4), 1007-1025, doi: 10.5194/hess-12-1007-2008.

Li, L., L. Zhang, H. Wang, J. Wang, J. Yang, D. Jiang, J. Li, and D. Qin (2007), Assessing the impact of climate variability and human activities on streamflow from the Wuding River basin in China, Hydrol. Processes, 21(25), 3485-3491, doi: doi:10.1002/hyp.6485.

Liu, X., Q. Tang, H. Cui, M. Mengfei, D. Gerten, S. Gosling, Y. Masaki, Y. Satoh, and Y. Wada (2017), Multimodel uncertainty changes in simulated river flows induced by human impact parameterizations, Environ. Res. Lett., 12(2), 025009, doi: 10.1088/1748-9326/aa5a3a.

Müller Schmied, H., S. Eisner, D. Franz, M. Wattenbach, F. T. Portmann, M. Flörke, and P. Döll (2014), Sensitivity of simulated global-scale freshwater fluxes and storages to input data, hydrological model structure, human water use and calibration, Hydrol. Earth Syst. Sci., 18(9), 3511-3538, doi: 10.5194/hess-18-3511-2014.

Pokhrel, Y. N., S. Koirala, P. J. F. Yeh, N. Hanasaki, L. Longuevergne, S. Kanae, and T. Oki (2015), Incorporation of groundwater pumping in a global Land Surface Model with the representation of human impacts, Water Resour. Res., 51(1), 78-96, doi: 10.1002/2014WR015602.

Siebert, S., J. Burke, J. M. Faures, K. Frenken, J. Hoogeveen, P. Döll, and F. T. Portmann (2010), Groundwater use for irrigation – a global inventory, Hydrol. Earth Syst. Sci., 14(10), 1863-1880, doi: 10.5194/hess-14-1863-2010.

Veldkamp, T. I. E., et al. (2018), Human impact parameterizations in global hydrological models improve estimates of monthly discharges and hydrological extremes: a multi-model

validation study, Environ. Res. Lett., 13(5), 055008, doi: 10.1088/1748-9326/aab96f.

Wada, Y., D. Wisser, and M. F. P. Bierkens (2014), Global modeling of withdrawal, allocation and consumptive use of surface water and groundwater resources, Earth Syst. Dynam., 5(1), 15-40, doi: 10.5194/esd-5-15-2014.

Wada, Y. (2016), Modeling Groundwater Depletion at Regional and Global Scales: Present State and Future Prospects, Surv. Geophys., 37(2), 419-451, doi: 10.1007/s10712-015-9347-x.

Wada, Y., et al. (2017), Human–water interface in hydrological modelling: current status and future directions, Hydrol. Earth Syst. Sci., 21(8), 4169-4193, doi: 10.5194/hess-21-4169-2017.

Yuan, X., Y. Jiao, D. Yang, and H. Lei (2018), Reconciling the attribution of changes in streamflow extremes from a hydroclimate perspective, Water Resour. Res., doi: doi:10.1029/2018WR022714.

Zhang, C., Q. Tang, and D. Chen (2017), Recent Changes in the Moisture Source of Precipitation over the Tibetan Plateau, J. Climate, 30(5), 1807-1819, doi: 10.1175/jcli-d-15-0842.1.

---

## Author Response (AR1)

We thank the editor and reviewers for their valuable comments. We have replied the comments one by one below, and revised the manuscript accordingly. The replies are highlighted in blue color and the modified texts (in the revised manuscript) are shown italic.

The Editor's comment

I have received two reviews of your manuscript. As you can see, both of them are very positive. But I would like to recommend for addressing the uncertainty in streamflow simulation, e.g., excluding those with NSE<0 simulations to see whether it affects main conclusions?

Reply: Thanks for the suggestion. Please note that the station data are only used for evaluating the multimodel simulations. We have re-calculated the average streamflow of stations over individual basins and China by excluding those with NSE<0 and redrawn the related figures (see Figures R1, R2 and R3). The inner plot in Figure R1, showing the observed and simulated streamflow seasonality of stations with NSE>0 (see Table S2) over China, has very little change compared to Figure 1 in the revised manuscript. In several northern basins (especially the Liao River), the seasonal streamflow (Figure R2) and annual streamflow (Figure R3) of the stations shows some differences from Figure S2 and S3 (in the revised manuscript). That is, the simulations and observations of the selected stations (NSE>0) are closer in these basins. Generally, the excluding of those stations do not have large effects because the excluded streamflows are relatively small. As the stations with NSE<0 are located in the northern basins (see Figure S2 below), the redrawn plots for the southern basins have no change.

In this study, we would like to use all station data for the evaluation. Excluding those stations with NSE<0 only affects the evaluation result, and would not affect the conclusions about the human and climate impacts on streamflow in China.

[Figure]

Figure R1. Multimodel medians of mean annual streamflow (MAF) in China from the VARSOC experiment. MAF medians are computed across 18 GHM-GMF combinations over the 1971-2000 period. The ensemble spread is represented by the ratio of interquartile range (IQR, 75th percentile minus 25th percentile) to the ensemble median of MAF (Median). The red circles indicate hydrological stations. The inner plot shows the comparison of the simulated seasonal streamflow (each GHM has three lines for the three GMFs) from the VARSOC experiment against the observations averaged for the hydrological stations with NSE>0 (see Table S2) over the period 1971-2000. The GHM names and basin names are the same as Figure 1 in the revised manuscript.

[Figure]

Figure R2. Seasonal cycle of streamflow from observations (orange) and multimodel medians (black). The observations are the average values of the hydrological stations with NSE > 0 (see Table S2), while the simulations are averaged values over the grid cells identified by the location of stations. The grey areas show the 25th and 75th percentiles of the multimodel simulations. Northern basins: Songhua River (SH), Liao River (LR), Northwest Rivers (NW), Hai River (HA), Yellow River (YR), Huai River (HU); Southern basins: Yangtze River (YZ), Southeast Rivers (SE), Southwest Rivers (SW), Pearl River (PR).

[Figure]

Figure R3. Simulated (black) and observed (orange) mean annual streamflow at the hydrological stations in each basin. The observations are the average values of the hydrological stations with NSE>0 (see Table S2), while the simulations are averaged values over the grid cells identified by the location of stations. The gray areas show the 25th and 75th percentiles of the multimodel simulations.

Reviewer #1

This study quantitative assessed the human impact and climate change impact on streamflow in continental China. The simulations streamflow was used from six global hydrological models driven by three meteorological forcings. The research is very interesting and significative. However, there are a few issues that the authors need to address before the manuscript can be accepted. I recommend most of the issues I raise below just need clarification or justification.
Reply: Thanks for the positive comment. We have replied the comments below and revised manuscript accordingly.

1. The simulated results need to be verified further with observed streamflow, maybe, QQPLOT, NSE etc. method can be used.
Reply: We have calculated NSE for the 44 stations, and added a sentence describing the result with a table (new added Table S2) in the revised supplementary information.

Revision in the manuscript (Subsection 3.1, the second paragraph (new added)):
*The Nash-Sutcliffe coefficients calculated for the multimodel median and observed monthly streamflow at each station (see Table S2) show that the multimodel medians have better performance in the southern basins.*

*Table S2. The Nash-Sutcliffe coefficients (NSE) for the simulated monthly streamflow from VARSOC experiment and observed monthly streamflow ($m^3$ $s^{-1}$) at the 44 stations over the 1971-2000 period. The observed mean annual streamflow (MAF, $m^3$ $s^{-1}$) averaged over the period is also shown for each station.*

| Number | Station Name | MAF | NSE | River name | Number | Station Name | MAF | NSE | River name |
|--------|-------------|------|-------|-------------------|--------|-------------|----------|-------|------------------|
| 1 | Guchengzi | 151.26 | -0.27 | Songhua River | 23 | Xixian | 117.87 | 0.31 | Huai River |
| 2 | Fuyu | 449.68 | 0.53 | Songhua River | 24 | Fuyang | 117.74 | 0.63 | Huai River |
| 3 | Tonghe | 1444.43 | 0.81 | Songhua River | 25 | Lutaizi | 639.00 | 0.80 | Huai River |
| 4 | Kuerbin | 26.94 | 0.004 | Songhua River | 26 | Bengbu | 800.63 | 0.81 | Huai River |
| 5 | Chaoyang | 18.00 | -0.88 | Liao River | 27 | Shishang | 1968.28 | 0.93 | Yangtze River |
| 6 | Chifeng | 7.76 | -0.25 | Liao River | 28 | Changyang | 431.02 | 0.75 | Yangtze River |
| 7 | Tieling | 84.36 | <-1.0 | Liao River | 29 | Pingshan | 4546.38 | 0.77 | Yangtze River |
| 8 | Liaozhong | 101.43 | 0.50 | Liao River | 30 | Sinan | 910.98 | 0.79 | Yangtze River |
| 9 | Changmapu | 29.30 | -0.37 | Northwest Rivers | 31 | Cuntan | 10747.92 | 0.68 | Yangtze River |
| 10 | Yingluoxia | 51.06 | -0.26 | Northwest Rivers | 32 | Datong | 28460.19 | 0.78 | Yangtze River |
| 11 | Zhamashenke | 22.70 | 0.09 | Northwest Rivers | 33 | Quzhou | 207.66 | 0.80 | Southeast Rivers |
| 12 | Sandaohezi | 16.45 | <-1.0 | Hai River | 34 | Zhuji | 40.24 | 0.58 | Southeast Rivers |
| 13 | Panjiakou | 60.87 | 0.01 | Hai River | 35 | Zhuqi | 1721.14 | 0.91 | Southeast Rivers |
| 14 | Luanxian | 96.09 | 0.71 | Hai River | 36 | Yangkou | 442.85 | 0.72 | Southeast Rivers |
| 15 | Xiapu | 4.58 | <-1.0 | Hai River | 37 | Daojieba | 1746.97 | 0.12 | Southwest Rivers |
| 16 | Huangbizhuang | 32.14 | -0.07 | Hai River | 38 | Gulaohe | 96.63 | 0.22 | Southwest Rivers |
| 17 | Cetian | 4.78 | -0.01 | Hai River | 39 | Manhao | 310.84 | 0.82 | Southwest Rivers |
| 18 | Lanzhou | 976.80 | 0.53 | Yellow River | 40 | Jiangbianjie | 194.96 | 0.68 | Pearl River |
| 19 | Shizuishan | 867.25 | 0.45 | Yellow River | 41 | Duanzhan | 2005.11 | 0.88 | Pearl River |
| 20 | Longmen | 803.67 | -0.47 | Yellow River | 42 | Xiayan | 449.63 | 0.82 | Pearl River |
| 21 | Huayuankou | 1103.51 | 0.09 | Yellow River | 43 | Wuxuan | 4130.25 | 0.81 | Pearl River |
| 22 | Xianyang | 107.26 | 0.63 | Yellow River | 44 | Boluo | 782.04 | 0.80 | Pearl River |

2. The simulated results are very bad in some basins, such as NW, SW, HA. These simulated streamflow need be post-processed, and then be used to analyzed the impact of human and climate change.

Reply: We recognized the poor performance of the simulations, especially in the northern basins (see above Table S2). A post-processing on the simulations could reduce the deviation in simulated streamflow from observations and narrow the spread across models (e.g., Yin et al., 2017). However, in this study, there are only limited stations (e.g., three stations in the Northwest Rivers and two stations in the Southwest Rivers) which cover small areas. If the limited number of the stations were used to correct the whole Northwest or Southwest regions, we found it would lead to very unrealistic streamflow estimates over rivers that we do not have streamflow observations (but know the mean annual streamflow from the reported statistics). Furthermore, the estimated water withdrawals in the models may be affected by the streamflow estimates. We would like to keep it consistent with water withdrawal estimates by the models. Due to the above reasons, we decided not to post-process the streamflow estimates. We have added a caution for the limited representative of the observations in the evaluation result to remind readers to treat it carefully.

Revision in the manuscript (Subsection 3.1, the second paragraph (new added)):
It should be noted that the stations are located at different reaches of individual basins. *Thus, the station-averaged estimates are largely dominated by those with large streamflow (e.g., at the lower reaches). Additionally, the coverage of stations used is relatively small (due to data availability), especially in hydrologically variable regions like in the Northwest Rivers, leading to not necessarily representative evaluation of the performance of the GHMs in the whole basin.*

3. The authors need add some explanation of ISIMIP2a about how to simulate water withdrawals.
Reply: We have added some description for the simulated water withdrawals in section 2.1 Simulation data.

Revision in the manuscript (Subsection 2.1, the second paragraph (new added)):
*Human impact considered in the VARSOC experiment (see the maps in Figure S1 and Table S1 for more details) includes the time-varying areas for both irrigated and rainfed cropland (Fader et al., 2010; Portmann et al., 2010) and reservoirs (dams) from the Global Reservoir and Dam (GRanD) Database (Lehner et al., 2011) including their commissioning year (see Figure S1 and Table S1 for more detail). Reservoir regulation was considered in the VARSOC experiment, which often reduces high streamflow in high-flow seasons and increases streamflow in dry seasons (Masaki, et al., 2017). Inter-basin water transfer was not considered in any of the model runs. The simulations of water withdrawals are different between the GHMs with respect to water use requirements and water withdrawal sources which are shown in Table S1. The sources of water withdrawals, depending on models, may include river channel, reservoirs, groundwater and lakes, and their fractions can be determined from reported statistics (e.g., Siebert et al., 2010) or estimated in models (Wada et al., 2014). In addition to the irrigation water requirement which is usually estimated by coupling crop models, most*

*GHMs considered the requirements for domestic and industrial water use which were prescribed in H08 (Hanasaki et al., 2008), LPJmL and MATSIRO (Pokhrel et al., 2015) or were estimated according to the population, socioeconomic and technological development in PCR-GLOBWB (Wada et al., 2014) and the population, thermal electricity production, gross added value, and technological change in WaterGAP (Flörke et al., 2013). Water use requirement for livestock was also prescribed in the LPJmL model, and estimated according to livestock densities in PCR-GLOBWB and WaterGAP2.*

Reviewer #2

The paper describes a multimodel assessment of the relative impacts of human activities and climate on mean annual streamflow over the past 4 decades in China. This study shows that unlike previous assessments, the climate impact signal is much more pronounced than the human impact signal in 88% of river segments in China. The study also quantifies the impact of humans across basins and discusses regional differences. In general the paper is publishable after some moderate revisions.

Reply: Thanks for the positive comments. We have replied the comments and revised the manuscript accordingly.

- The use of the term 'climate change' in the title and throughout the manuscript is somewhat confusing and misleading because it gives the impression that the paper will be forward looking in time and over the coming several decades (e.g., 2050, 2100). A more appropriate term is 'climate impacts'

Reply: We used "climate impact" in the title. We have replaced "climate change" by "climate variability" in most cases, and by "climate impact" in some cases where appropriate. "climate change" is still used when it refers to future climate change and results from some specific references.

- The 3rd paragraph of the introduction makes the argument that "This is the first study to perform such a quantitative assessment for all rivers of China with comparable modeling experiments." Being aware of the ISIMIP publications (https://www.isimip.org/outcomes/publications/) in this space with global assessments including many of the authors on this paper, I find this argument to be an exaggeration. I think the last sentence of that paragraph is a key novelty of this work, and as such linking back to the content of the second paragraph in the introduction to make the case would be my suggestion. I do agree that focusing on China is somewhat unique about this study. So one suggestion is to tweak the noted sentence as follow "This is the first study to focus on performing such a quantitative assessment for all rivers of China with comparable modeling experiments."

o Schewe et al.: Multimodel assessment of water scarcity under climate change. PNAS, 2014.

o Haddeland et al.: Global water resources affected by human interventions and climate change, P. Natl. Acad. Sci. USA, 111, 3251–3256, https://doi.org/10.1073/pnas.1222475110, 2014.

o Veldkamp et al.: Water scarcity hotspots travel downstream due to human interventions in the 20th and 21st century, Nature Commun., 8, 15697, https://doi.org/10.1038/ncomms15697, 2017.

o Wada et al.: Human–water interface in hydrological modelling: current status and future directions, Hydrol. Earth Syst. Sci., 21, 4169-4193.

Reply: Thanks for the suggestion. We have revised the sentence following the suggestion.

- P5, L2: I would suggest omitting 'preliminary'

Reply: Changed.

- P7, L14-17: Showing the individual models in figure S2 makes the figure too busy to read. Why not use the same format as in figure 2 by showing a band around the median. Also, it would be useful to show the same type of figure as figure 2 but for streamflow.

Reply: We have redrawn Figure S2 and added a figure (Figure S3 in the revised manuscript and below) for the simulated and observed annual streamflow following the suggestion.

A brief description has been added.

Revision in the manuscript (Subsection 3.1, the second paragraph (new added)):
*The model spreads in the ensembles of seasonal streamflow and the deviations between observation and simulations are relatively larger in the northern basins than those in the southern basins (see Figure S2 for each basin). Comparison between the simulated and observed annual streamflow (Figure S3) shows similar patterns as the seasonal streamflow with respect the discrepancies between northern and southern basins. The Nash-Sutcliffe coefficient was calculated for the multimodel median and observed monthly streamflow at each station (see Table S2), which shows that the multimodel medians have better performance in the southern basins. This evaluation indicates that the multimodel simulations have relatively poor performance in northern basins and most stations with low Nash-Sutcliffe coefficients have smaller streamflow (e.g., in dry areas or upper reaches). The large spreads between models underline the necessity of using ensemble medians rather than individual models for the attribution of streamflow changes.*

[Figure]

*Figure S2. Seasonal cycle of streamflow from observations (orange) and multimodel medians (black). The observations are the average values of the hydrological stations, while the simulations are averaged values over the grid cells identified by the location of stations. The grey areas show the 25th and 75th percentiles of the multimodel simulations. Northern basins: Songhua River (SH), Liao River (LR), Northwest Rivers (NW), Hai River (HA), Yellow River (YR), Huai River (HU); Southern basins: Yangtze River (YZ), Southeast Rivers (SE), Southwest Rivers (SW), Pearl River (PR).*

[Figure]

*Figure S3. Simulated (black) and observed (orange) mean annual streamflow at the hydrological stations in each basin. The observations are the average values of the hydrological stations, while the simulations are averaged values over the grid cells identified by the location of stations. The grey areas show the 25th and 75th percentiles of the multimodel simulations.*

- P7, L18-24: I realize that given the large departures in water withdrawal estimates, matching streamflow gauge observations might be a challenge, unless the authors believe that simulated water withdrawals might be equally or even more reliable than the statistically collected data, which have their own challenges.

Reply: We would not state that the simulations of water withdrawals are equally or more reliable than the statistically collected data. In fact, they are largely based on statistically collected data (e.g., Flörke et al. 2013; Hanasaki et al., 2008). We agree that the large spreads in the multimodel simulations of water withdrawals should be one uncertainty source to the streamflow simulations, but the effect is not superimposing (e.g., Fig. 9 in Müller Schmied et al 2014 for one GHM). On the other hand, Veldkamp et al. (2018) showed that inclusion of human impacts such as water withdrawals leads to better model performances. The simulations of water withdrawals remain a challenge, though great efforts have been made by the community. We have added a statement to address this concern at the end of this paragraph in the revised manuscript, but would not further elaborate it since it is not the focus of this study.

Revision in the manuscript (Subsection 3.1, the last paragraph)
*The large deviations in the multimodel simulations of water withdrawals could make the modeling of streamflow more challenging (Döll et al., 2016; Wada et al., 2017).*

- P7, L18-24: Are water withdrawals taken from surface water sources or also groundwater sources? What about return flows? Also, I am assuming that glacier melting, which contributes to streamflow, is simulated in these models, but that region is not included in the analysis. I realize that some of these were mentioned in the results, but incorporating some of these details briefly when discussing the method or the results from the evaluation exercise would suffice.

Reply: The sources of water withdrawals are shown in Table S1, depending on models, which may include river channel, reservoirs, groundwater and lakes. Return flows were considered in different ways for different water uses (e.g., Müller Schmied et al., 2014; Wada et al., 2014; Pokhrel et al., 2015). Glacier melting was not simulated in most GHMs (except PCR-GLOBWB) in this study. We have added the absence of glacier melting in the models as a reason for excluding the Tibet plateau region, and described the sources of water withdrawals in the Method section (please see the reply to the 3rd comment of reviewer #1) in the revised manuscript.

Revision in the manuscript (Subsection 2.1, the first paragraph)
The simulations may have large uncertainties over the Tibetan Plateau because long-term meteorological and streamflow observations are sparse in this region (Zhang et al., 2017) *and the modeling of glacier melting is absence in most of the models*.

- P10, L23-29: how does the model specify how much water is taken from surface water vs groundwater sources? Are the small pockets of increased MAF due to human impacts (Fig 5c) attributed to technological change (e.g., irrigation efficiency), or return flow from groundwater pumping, or something else?

Reply: Generally, groundwater is withdrawn when the water use requirement is not met due to limited accessibility to or insufficient quality of surface water. Groundwater withdrawal was considered in most models (see Table S1), but the pumping rate may vary substantially

between models (Wada et al., 2016). It is difficult to determine the groundwater pumping rate since groundwater storage is usually unknown. The fraction of groundwater for water use is determined from reported statistics data (Siebert et al., 2010, used in WaterGAP) or estimated in the model (e.g., PCR-GLOBWB, see Wada et al., 2014). We have briefly clarified it in the Method section (also see the reply to the 3rd comment of reviewer #1) in the revised manuscript.

The increased MAF should be mainly due to return flow, but we cannot identify it from which source because of lacking related model output currently. Technological development may improve water use efficiency and reduce the amount of withdrawals. However, it may be not the reason for the slight MAF increase induced by DHI change, because water withdrawal increased over the study period (see Figure 2). We have clarified it in the revised manuscript.

Revision in the manuscript (Subsection 3.4, the first paragraph)
Compared to the first sub-period, in the second sub-period MAF increased by more than 30% in many river segments of the Northwest Rivers and increased by more than 5% in large parts of the Huai River, *which may be due to the return flow from water withdrawals.*

- P11, L7: I would suggest omitting the sentence about the US and Canada. It breaks the flow of the paragraph which is talking specifically about China.
Reply: Removed.

- P13, L14-30: To me this, this is a key contribution of this study. Yes, I agree that the results are not necessarily comparable in term magnitudes due to the highlighted reasons by the authors. But a missing discussion point is to why they fundamentally differ in their findings. I don't agree that either one of these two approaches (small scale using statistical approaches vs large scale modeling similar to this study) is necessarily superior. Each approach has its own pros and cons. So articulating why this approach differs from earlier findings is critical.
Reply: We agree that both the methods has its own pros and cons. At the end of the discussion, we have emphasized the importance of using multiple approach to obtain more reliable assessment. In the revised manuscript, we have rewritten the last paragraph of the discussion, wherein we clearly stated the key difference between the method in previous studies and this study.

Revision in the manuscript (Subsection 4.6, the last paragraph)
*One major difference between previous studies (e.g., Li et al., 2007; Bao et al., 2012) and this study is that the former estimates DHI contribution by comparing simulations with observations while we compare two simulation experiments. The former may be subject to uncertainty in comparing the data from two systems (i.e., the model and the real world). In this study, the two simulation experiments favor the estimation of DHI contribution in a consistent manner that is largely free of uncertainty in the data from different systems. The multimodel simulations also allow profiling the uncertainties among models and input forcings, which is*

*difficult for a single model assessment. However, the deficiency of this approach is that DHI is not real. Therefore, the assessment is inevitably influenced by the extent to which the models can reproduce the real DHI.* Considering the complexity of DHI on streamflow and the ability of current hydrological models in reproducing historical hydrological changes, multimodel simulations and different attribution approaches are well worth obtaining more robust assessments (Liu et al., 2017; Yuan et al., 2018).

[revised manuscript text omitted]